# Molecular surveillance of insecticide resistance in *Phlebotomus argentipes* targeted by indoor residual spraying for visceral leishmaniasis elimination in India

Emma Reid[1‡], Rinki Michelle Deb[1‡], Asgar Ali[2], Rudra Pratap Singh[1], Prabhas Kumar Mishra[1], Josephine Shepherd[1], Anand Mohan Singh[2], Aakanksha Bharti[2], Chandramani Singh[3], Sadhana Sharma[3], Michael Coleman[1‡*], David Weetman[1‡]

**1** Liverpool School of Tropical Medicine, Liverpool, United Kingdom, **2** CARE India, Patna, India, **3** All India Institute of Medical Sciences, Patna, India

‡ EM and RMD share first authorship on this work. MC and DW are joint senior authors on this work.
* Michael.Coleman@lstmed.ac.uk

**Data Availability Statement:** Data is contained in the manuscript and supporting files.

## Abstract

Molecular surveillance of resistance is an increasingly important part of vector borne disease control programmes that utilise insecticides. The visceral leishmaniasis (VL) elimination programme in India uses indoor residual spraying (IRS) with the pyrethroid, alpha-cypermethrin to control *Phlebotomus argentipes* the vector of *Leishmania donovani*, the causative agent of VL. Prior long-term use of DDT may have selected for knockdown resistance (*kdr*) mutants (1014F and S) at the shared DDT and pyrethroid target site, which are common in India and can also cause pyrethroid cross-resistance. We monitored the frequency of these marker mutations over five years from 2017–2021 in sentinel sites in eight districts of north-eastern India covered by IRS. Frequencies varied markedly among the districts, though finer scale variation, among villages within districts, was limited. A pronounced and highly significant increase in resistance-associated genotypes occurred between 2017 and 2018, but with relative stability thereafter, and some reversion toward more susceptible genotypes in 2021. Analyses linked IRS with mutant frequencies suggesting an advantage to more resistant genotypes, especially when pyrethroid was under-sprayed in IRS. However, this advantage did not translate into sustained allele frequency changes over the study period, potentially because of a relatively greater net advantage under field conditions for a wild-type/mutant genotype than projected from laboratory studies and/or high costs of the most resistant genotype. Further work is required to improve calibration of each 1014 genotype with resistance, preferably using operationally relevant measures. The lack of change in resistance mechanism over the span of the study period, coupled with available bioassay data suggesting susceptibility, suggests that resistance has yet to emerge despite intensive IRS. Nevertheless, the advantage of resistance-associated genotypes with IRS and under spraying, suggest that measures to continue monitoring and improvement of spray quality are vital, and consideration of future alternatives to pyrethroids for IRS would be advisable.

**Funding:** This work was supported in whole by the Bill & Melinda Gates Foundation OPP1151797 to MC. Under the grant conditions of the Foundation, a Creative Commons Attribution 4.0 Generic License has already been assigned to the Author Accepted Manuscript version that might arise from this submission. The funders had no role in study design, data collection and analysis, decision to publish, or preparation of the manuscript.

**Competing interests:** The authors have declared that no competing interests exist.

## Author summary

Visceral leishmaniasis (VL) is a deadly parasitic disease with a primary focus in north-eastern India. Control of the sand fly, *Phlebotomus argentipes*, vector of VL in India, is primarily reliant upon spraying the internal walls of houses and animal shelters with residual pyrethroid insecticide. Spray programmes depend upon well-controlled spraying and effective insecticides to which the targeted insects are susceptible. Changing insecticides is logistically challenging, therefore early detection of insecticide resistance is crucial. As part of a wider programme of entomological surveillance we used molecular resistance assays of knockdown resistance (*kdr*) mutations to investigate evidence for changing resistance profiles, and possible links with the spraying programme across a system of eight districts in north-eastern India. Mutant frequencies varied substantially in space and time, with a major change across the first two years of the study, but stability for the remainder. Resistance-associated *kdr* alleles were positively associated with indices of spray coverage and with under spraying, suggesting that this creates vulnerability to development of pyrethroid resistance. However, the most strongly resistance conferring mutant genotype was rarely detected, suggesting overall that notable resistance is not yet emerging, despite wide coverage of the spray programme. This is an encouraging result for the VL elimination programme but with apparent advantage of resistance alleles in sprayed areas it would be wise to seek alternative insecticides for spraying.

## Introduction

Between 2004 to 2010 there were an estimated 200,000 to 400,000 cases and 50,000 deaths annually of visceral leishmaniasis (VL), also known as Kala-azar, making this the second deadliest parasite after malaria. Currently 130 million people in India from four states, Bihar, Jharkhand, Uttar Pradesh and West Bengal are at risk from VL, however, only 3145 cases were recorded in 2019 in India reflecting the success of the elimination programme [1]. In India VL is caused by the parasite *Leishmania donovani*, transmitted solely by the sand fly *Phlebotomus argentipes*.

Vector-based control of VL was originally a by-product of IRS campaigns using dichlorodiphenyltrichloroethane (DDT) from the National Malaria Eradication Programme in the 1960s and 1970s. However, with reduction of anti-malaria IRS campaigns in the 1970s VL cases began to rise again [2,3]. In 2005, a tripartite agreement between Bangladesh, India, and Nepal was signed with the aim of eliminating VL and post–kala-azar dermal leishmaniasis as a public health problem *i.e.* to less than one case per 10,000 population by 2015, this was extended to 2020 [4], and recently extended to 2030 due to effects of the Covid-19 pandemic. India is on target to achieve elimination as rates of VL decline are currently at their lowest ever levels [5]. Current measures employed in the VL elimination efforts in India include early case detection with effective treatment, surveillance, and vector control with IRS. Historically DDT was used for IRS, and was used initially in the elimination campaign from 2005, however prompted in part by resistance in local *P. argentipes* populations to DDT [6] a change was made in 2015 and 2016 to spraying with the pyrethroid alpha-cypermethrin.

DDT and pyrethroids share the same mode of action, both binding to the voltage gated sodium channel (Vgsc) resulting in repetitive nerve firing, paralysis and death of the insect [7,8]. Multiple mutations in the Vgsc gene cause DDT and pyrethroid 'knockdown resistance' (*kdr*) in insects, the most common occurring at codon 1014 (using *Musca domestica* codon

numbering) [9]. Partial sequencing of the Vgsc in *P. argentipes* from Bihar detected two *kdr* mutations, which change the wild-type, insecticide susceptible leucine allele at codon 1014 to either phenylalanine (L1014F) via either of two nucleotide changes, or serine (L1014S) [8]. Both amino acid mutations were significantly elevated in first-lab-generation (F1) female *P. argentipes* surviving exposure in tube bioassays using *Anopheles* diagnostic doses of DDT or reduced duration assays of alpha-cypermethrin and deltamethrin. Though 1014F confers somewhat stronger DDT-resistance and pyrethroid-tolerance than 1014S, two mutant copies (FF, FS or SS) appeared to be required for a resistant phenotype in most cases, giving a pragmatic binary separation into *kdr* and non-*kdr* genotypes, which displayed strong predictive value as resistance markers [8]. Other studies have also established an association between *P. argentipes* survivorship in pyrethroid bioassays in the laboratory [10,11], though possession of *kdr* mutations does not necessarily equate to survival [12].

Unlike for DDT, potentially operationally relevant pyrethroid resistance has yet to be detected in *P. argentipes* [13] but monitoring for signals of changes in resistance is an essential part of control programmes. Owing to their lab-intractability arising from a relatively prolonged life cycle and sensitivity to rearing conditions, broad-scale phenotypic assessment of insecticide resistance is challenging in *P. argentipes*. In addition, approved diagnostic doses to detect deviation from susceptibility have only just become available for sand flies [14]. In contrast, molecular surveillance using sensitive and specific DNA assays targeting the L1014F and S *kdr* mutations provide a high-throughput tool for spatial and temporal monitoring to detect changes indicative of shifting resistance profiles. Whilst the 1014F and S mutations alone may not lead to high level pyrethroid resistance, experience from African *Anopheles* shows how prolonged pyrethroid exposure in wild populations leads to accumulation of supplementary *kdr* mutations [15], as well as addition of metabolic resistance mutants [16]. Moreover, though IRS spraying quality in Bihar has improved markedly, variation inevitably remains [13], which *P. argentipes* possessing resistance mechanisms may be able to exploit.

We report results from wide-scale spatio-temporal molecular surveillance of *kdr* mutations in *P. argentipes* sampled from sentinel sites spanning the most VL-endemic areas of north-eastern India. The primary aims of the study were to determine whether changes in resistance marker frequency have occurred during the period of intensive alpha-cypermethrin spraying, and whether any variation might be linked to, or represent a future risk for, the IRS programme. Whilst a marked shift in *kdr* genotypes and mutant alleles was seen across multiple sentinel sites between the first and second years of sampling (2017–18), frequencies thereafter remained stable to 2021. This stability indicates that, despite evidence linking IRS activities to advantages for resistant genotypes, overall IRS does not currently appear to be selecting for enhanced resistance across the study area.

## Methods

### Sentinel sites

Eight sentinel sites in VL endemic areas were established in north-eastern India: six in Bihar, one in Jharkhand and one in West Bengal as described previously [13]. The districts are East Champaran, Darjeeling, Godda, Gopalganj, Katihar, Muzzafapur, Purnia and Samastipur (Fig 1). In brief, each site had at least 1 new VL case per 10,000 persons per year at sub-district (block) level. Block selection was based on total reported VL case numbers, extracted from the 2015 district level IRS micro-plan data. Villages were selected if they had a VL case history for the previous three consecutive years and appropriate infrastructure to allow year-round village access. Of the villages that met the selection criteria, between four and seven villages per sentinel site were selected using a random number generator in Microsoft Excel.

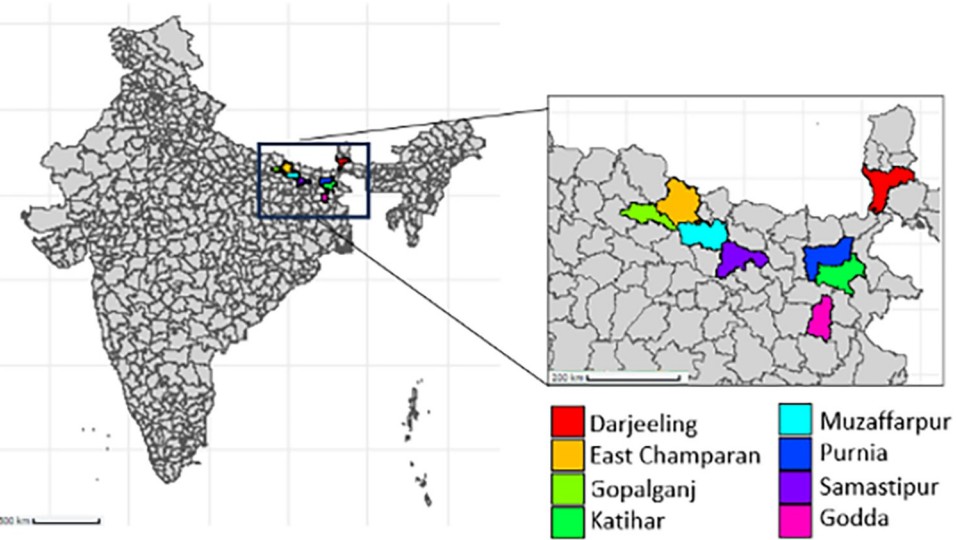

**Fig 1. Map displaying location of districts.** *The map was created using R with the baselines coming from GADM, the licence gives permission to use in academic articles (https://gadm.org/license.html)).*

### *Phlebotomus argentipes* collections

Year-round sand fly collections with CDC light traps were made from 15 randomly selected houses in each village within the eight districts over a period of two consecutive nights (6:00 PM to 6:00 AM) every two weeks [13] from 2017 to 2021. The light traps were hung in the corner of a bedroom and optimally positioned 15 cm away from the wall and 5 cm above ground. All sand flies were identified to species level by morphological criteria from established taxonomic keys [17] and *P. argentipes* stored in 1.5ml Eppendorf tubes over silica for further analysis.

### *Kdr* marker genotyping

DNA was extracted from female sand flies using STE buffer with proteinase K by heating on a thermocycler for 60 minutes at 65˚C then 15 minutes at 95˚C. Using a maximum number of five females per household, samples were randomly selected for each district. DNA was used as template for genotyping to detect *kdr* mutations *Vgsc*1014S and *Vgsc*1014F [8]. Two Taqman qPCRs were used to determine genotype, using TaqMan Gene Expression Master Mix and specific primers and probes (Thermofisher life sciences) developed by [8], following the same assay protocol. Thermal cycling conditions were as follows; 95˚C for 3 minutes followed by 40 cycles of 95˚C for 10 seconds and 55˚C for 30 seconds, results were analysed using CFX Maestro Software (Bio-Rad) and MxPro–Mx3005P Software (Agilent).

### IRS coverage and quality

Details of assessment of IRS coverage and quality assessment using HPLC are described in detail elsewhere [13]. For the analysis here we used the proportion of villages that were sprayed within 10km of a village from which sand flies were collected as an index of local spray coverage, and averaged results from the spray rounds in years where more than one occurred. For an index of spray quality, we used the proportion of houses sampled within a village determined to be under sprayed based on the classification in [13].

### Data analysis

Genotypes were scored according to their amino acids, and additionally according to whether the genotype might be predicted to lead to a knockdown resistance phenotype, assuming a recessive nature of the mutants [8]. Thus, individuals possessing two mutant alleles, whether S/S, S/F or F/F at position 1014 are considered '*kdr* genotypes', whilst those with either two wild type leucine alleles or heterozygotes (L/S or L/F) are considered non-*kdr* genotypes. Data were analysed using generalised linear models (GzLM) with a binomial logit link function in Stata 16, with village level analyses including district (= sentinel site) as a random effect to account for clustering. Mean changes in genotype or allele frequency from the first collection year were compared to starting frequencies at village and district levels using Spearman rank correlation in SPSS v26. Two analyses were performed to determine possible influences of IRS on genotype and allele frequencies. (i) Data from 2018–2020 (excluding Darjeeling which lacked IRS data), were analysed using GZLMs including district and year as before, but also including an index of alpha-cypermethrin spraying coverage (above) from the preceding year, and the proportion of houses determined to be under-sprayed (above) as covariates. Analysis was limited to 2018–2020 owing to availability of both IRS coverage and HPLC data for the years 2017–2019. (ii) Data from 2017 were compared between IRS and non-IRS villages within districts, with district included as a random variable in the GZLM. This analysis was only possible for 2017 because in subsequent years data from IRS and non-IRS villages were not available from within districts.

## Results

### Spatial and temporal variation in marker frequency

A total of 17,680 sand flies collected from 46 villages within the eight districts between 2017 and 2021 were successfully genotyped at the Vgsc 1014 locus. Results for the *kdr* genotype classification (i.e. genotypes with two mutant 1014 alleles) are illustrated in Fig 2; substantial variation among districts and collection years is evident, with each highly significant (P<0.001 for both district and year; Table 1). The major change in *kdr* genotype frequency occurred between 2017 and 2018 with stable frequency to 2020 and then a reduction in 2021 (Fig 2), though frequencies remained significantly higher than in 2017 (Table 1). This temporal change

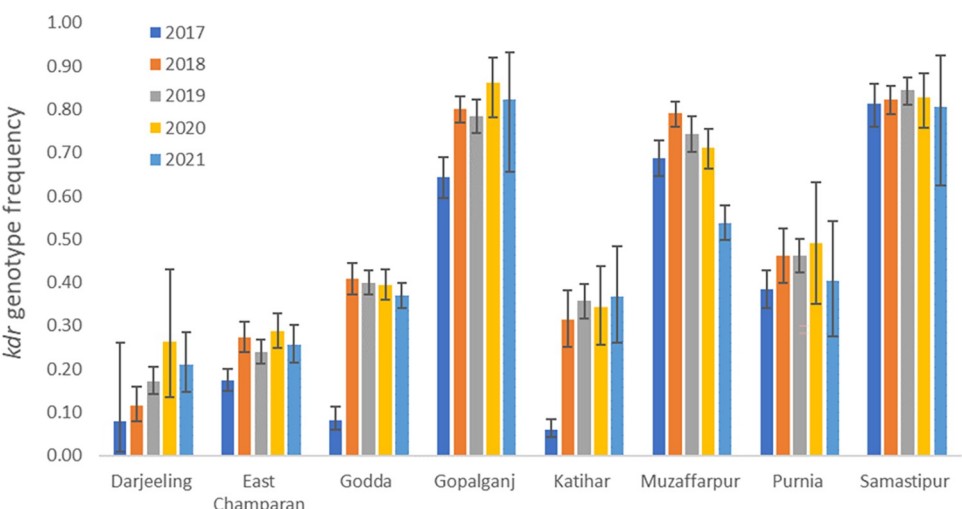

**Fig 2. *kdr* genotype frequencies (mean +/- 95% confidence intervals) across the 5-year period for the 8 districts.**

**Table 1. GZLM (binomial) analysis of predictors of *kdr* genotype frequency.**

| Source | odds ratio | 95% lower C.I. | 95% upper C.I. | P-value |
|---|---|---|---|---|
| Intercept | 2.716 | 2.337 | 3.156 | |
| *District* | | | | |
| Darjeeling | 0.037 | 0.030 | 0.045 | <0.001 |
| East Champaran | 0.067 | 0.057 | 0.079 | <0.001 |
| Godda | 0.114 | 0.098 | 0.133 | <0.001 |
| Gopalganj | 0.698 | 0.586 | 0.832 | <0.001 |
| Katihar | 0.074 | 0.062 | 0.089 | <0.001 |
| Muzaffarpur | 0.505 | 0.431 | 0.592 | <0.001 |
| Purnia | 0.175 | 0.148 | 0.207 | <0.001 |
| Samastipur (reference) | | | | |
| *Year* | | | | |
| 2017 (reference) | | | | |
| 2018 | 2.127 | 1.915 | 2.363 | <0.001 |
| 2019 | 2.040 | 1.846 | 2.254 | <0.001 |
| 2020 | 2.116 | 1.868 | 2.398 | <0.001 |
| 2021 | 1.539 | 1.361 | 1.741 | <0.001 |

in frequency from 2017 was significant in five of the eight districts (Table 2). Spatial variation among villages within districts was non-significant in most cases though evident among villages in Godda and Darjeeling (Table 2). In the latter, the spatial variation, combined with relatively small sample sizes may have obscured statistical detection of an apparent temporal pattern of change (Fig 2). There was a significant trend (Spearman's $\rho$ = -0.66, N = 33, P<0.001) for villages with lower *kdr* frequencies in 2017 to show a higher increase over subsequent years (Fig 3). Though this analysis did not account for clustering of villages, the pattern was also significant across districts when pooling village-level data (Spearman's $\rho$ = -0.80, N = 8, P = 0.018).

Temporal variation in the frequency of the three 1014 alleles reflected that of the *kdr* genotype, albeit in opposing direction for wild type leucine, with the same sharp change from 2017 to 2018 and stability thereafter. In 2021 whilst phenylalanine remained stable, the serine allele decreased in frequency and the leucine allele decreased relative to the 2017 starting point (S1 Table). The pattern of spatial variation among districts in the leucine allele was also very similar (again in opposing direction) to that observed for *kdr* genotypes. Spatial variation of the two resistant alleles was more variable, with the phenylalanine and serine interchanging in

**Table 2. Summary of GZLM results showing P-values for effects of village and year within each district on *kdr* genotype variation.**

| *District* | Village | Year | Pairwise differences |
|---|---|---|---|
| Darjeeling | <0.001 | NS | |
| East Champaran | NS | <0.001 | 2017<others |
| Godda | <0.001 | <0.001 | 2017<others |
| Gopalganj | NS | <0.001 | 2017<others |
| Katihar | NS | <0.001 | 2017<others |
| Muzaffarpur | NS | <0.001 | 2017<2018; 2021<others |
| Purnia | NS | NS | |
| Samastipur | NS | NS | |

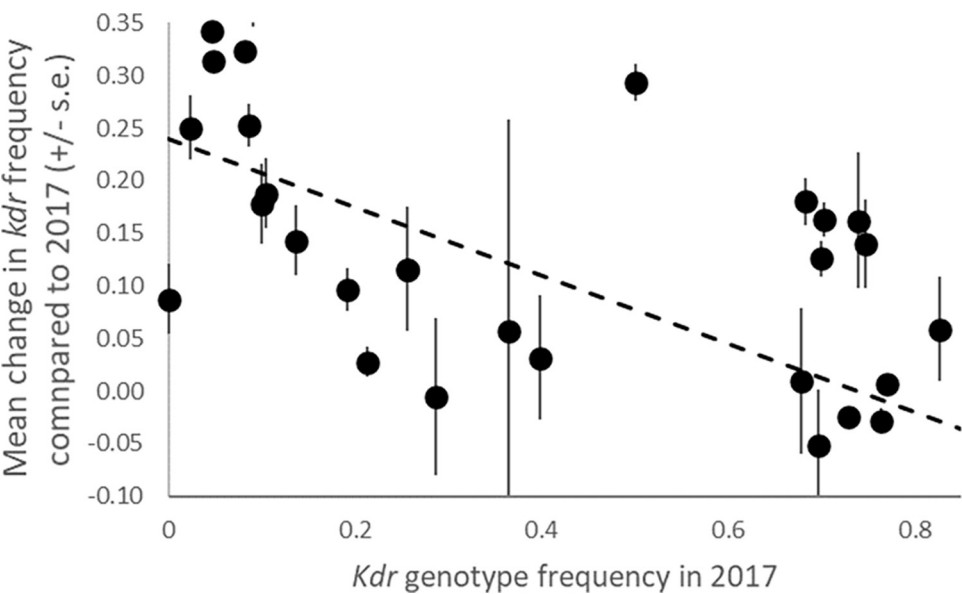

**Fig 3. Relationship between *kdr* genotype frequency in the first sampling year and average changes from the initial value over subsequent sampling years at the village level.** Dashed line showing linear regression fit is provided for visual illustration of trend.

frequency between relatively higher or lower levels (S1 Table) yielding parity in *kdr* genotypes (Table 1). Overall frequencies of each allele were remarkably balanced (L:S:F = 32%:37%:31%) with relatively little variation in proportions between the start and end point of the five years of collections (Fig 4). Major changes in frequencies of all but the serine homozygote (S/S) genotype occurred between 2017 and 2018, followed by stability over the subsequent years, with reductions in frequency of S/S and an increase in the wild type homozygote L/L in 2021, relative to 2017 (Fig 5).

## Association of IRS with 1014 marker variation

To investigate whether IRS might influence spatial and temporal variation in 1014 genotypes and allele frequencies we performed two analyses which aimed to test the hypothesis that insecticide pressure from IRS may favour certain alleles or genotypes. The first analysis

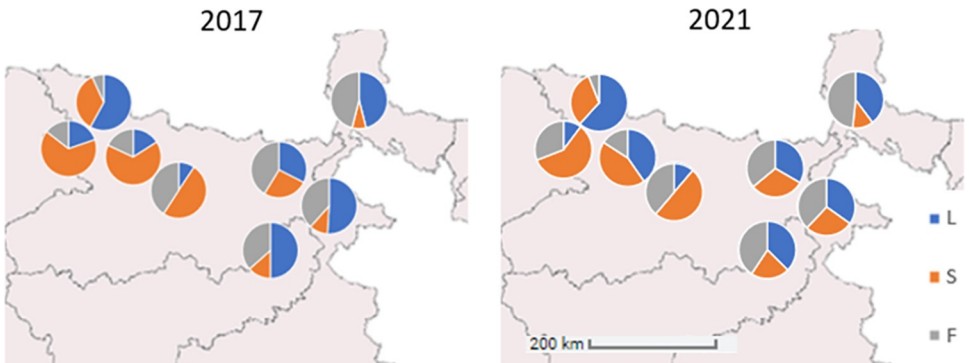

**Fig 4. Map comparisons of 2017 data and 2021 data showing the 1014 alleles.** *The map was created using R with the baselines coming from GADM, the licence gives permission to use in academic articles (https://gadm.org/license.html)).*

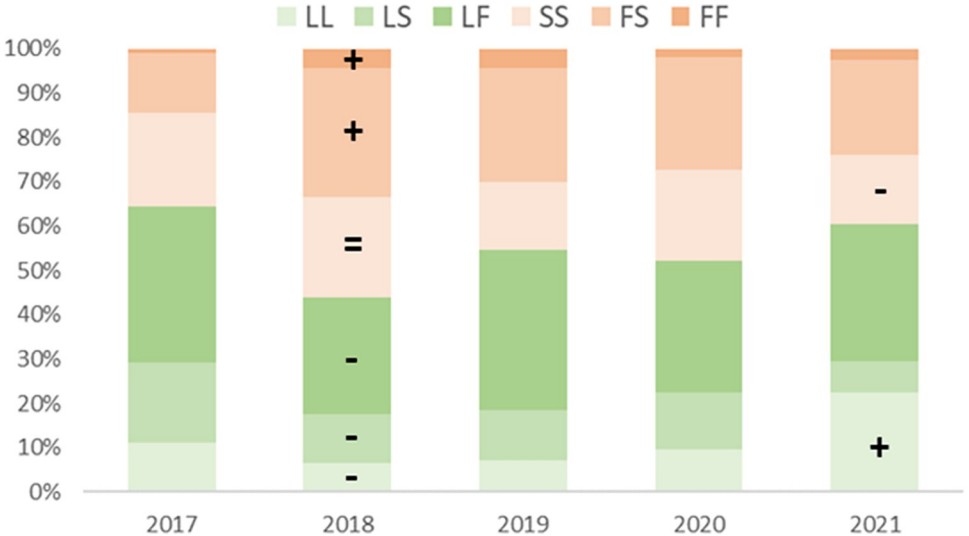

**Fig 5. Genotype proportions for the 1014 marker in each sampling year.** Comparison with the 2017 reference year is indicated by the +/-/ = signs in 2018 (see S1 Table for statistical test results); subsequent years show the same relationship with 2017 unless indicated by a different symbol.

involved a GZLM including district and year as factors (as in Table 1 above) but additionally included two covariates: the proportion of houses in surrounding villages covered by alpha-cypermethrin IRS and the proportion of houses classified as under sprayed by HPLC, each in the year preceding the sample collection. The model was limited to marker data from 2018–2020 for which both the preceding year's spray coverage and HPLC data were available. This analysis followed the reasoning that given available evidence for mutants' association with pyrethroid tolerance, rather than necessarily full resistance (see Introduction), selection could be influenced by both coverage and substandard dosing. The second analysis examined marker frequency differences between villages receiving IRS or not from within the same districts and was limited to 2017 for which such pairings within districts were available. Results are summarised in Table 3a with odds ratios greater than one indicating a positive relationship between

**Table 3. Summary of GZLM results (odds ratios and P-values) for: (a) effects of IRS coverage with alpha-cypermethrin in the previous year, with proportion of HPLC results indicating underspraying; (b) comparison of IRS vs non-IRS villages in 2017, on marker frequencies.**

|  | (a) IRS coverage | | (a) IRS underspraying | | (b) IRS vs not (2017) | |
|---|---|---|---|---|---|---|
| *genotype or allele* | Odds ratio | P-value | Odds ratio | P-value | Odds ratio | P-value |
| *kdr* | 1.49 | 0.003 | 1.13 | 0.20 | 0.99 | 0.92 |
| LL | 0.37 | <0.001 | 0.56 | 0.004 | 0.65 | 0.43 |
| LS | 0.35 | <0.001 | 0.56 | 0.001 | 0.78 | 0.05 |
| LF | 1.30 | 0.08 | 1.28 | 0.026 | 1.70 | 0.019 |
| SS | 1.09 | 0.58 | 1.15 | 0.30 | 0.79 | 0.16 |
| FS | 1.62 | <0.001 | 0.91 | 0.38 | 3.09 | 0.30 |
| FF | 0.72 | 0.40 | 1.93 | 0.013 | 0.72 | 0.77 |
| L | 0.69 | <0.001 | 0.86 | 0.042 | 0.9 | 0.26 |
| S | 1.04 | 0.63 | 0.95 | 0.52 | 0.83 | <0.001 |
| F | 1.40 | <0.001 | 1.16 | 0.048 | 1.36 | <0.001 |

models: (a) District, Year, proportionate SP-IRS in previous year, proportion of houses undersprayed; (b): IRS appled (yes/no) district (random variable).

genotype or allele frequency in a sampling year and proportionate IRS coverage in the preceding year; odds ratios less than one indicate a negative relationship.

Relationships between IRS coverage and marker frequencies broadly followed *a priori* expectations for resistance association. Frequencies of the leucine allele and two of the three leucine-containing genotypes (LL and LS) were significantly negative, with LF positive but not significant. Frequencies of the phenylalanine allele, the FS genotype and the *kdr* genotype group were significantly positively related to IRS coverage (Table 3A). Proportionate IRS under spraying showed similar effects on the frequencies of the leucine allele and genotypes, though here the positive association of LF frequency was significant (i.e., higher with under spraying). Phenylalanine allele and homozygote genotype frequencies were significantly positively related to IRS under spraying, whilst neither the *kdr* genotype group nor FS genotype were significantly associated (Table 3A). More limited data from the 2017 IRS vs no-IRS village comparison also support a link with the LF genotype and phenylalanine allele, each of which was significantly more common in IRS villages, as well as detection of a lower frequency of the serine allele, though again the *kdr* genotype group did not vary significantly (Table 3B).

It should be noted that there was a strong correlation between alpha-cypermethrin-IRS coverage and previous DDT-IRS coverage in 2014–2016 (S1 Fig) in the same groups of villages (r = 0.48–0.70, N = 43, P≤0.001 for each year between 2016 and 2020). Thus, whilst relationships between allele and genotype frequencies and alpha-cypermethrin-IRS pressure from the preceding year's coverage, might be consistent with relative advantage or disadvantage, an influence of older spraying coverage history cannot be discounted. Nevertheless, relationships with under spraying are also at least partially consistent with expectation of selection in combination with IRS pressure, in terms of negative relationships with the 'non-*kdr* genotypes' LL and LS, and a positive relationship with the *kdr*-linked genotype FF. However, for both results from under spraying and comparison of IRS and non-IRS villages, the significant positive relationship with LF (not expected to confer *kdr*) and lack of significance of the *kdr* genotype group do not meet *a priori* expectations.

To further investigate the relationship among genotypes and possible evidence for selection, Hardy-Weinberg (H-W) expectations were calculated for each district-year sample set (Fig 6). The majority of tests showed significant deviation of observed genotypes from expectations (33/40 P<0.05, following Bonferroni multiple testing correction), indicating widespread departure from H-W. Barring some exceptional, and relatively spatially variable results for SS and FS in 2017 (evident from high standard errors), the general pattern was of under-representation of LL, LS and FF genotypes, slight over-representation of SS and FS genotypes, and strong over-representation of the LF genotype. The latter is suggestive of a relative benefit of the LF genotype. Taken together with the results from the IRS-association analysis (Table 3) this may indicate a hitherto unexpected benefit of this wild-type/resistant allele heterozygote in the field populations surveyed when exposed to IRS.

## Discussion

India is making significant progress toward elimination of visceral leishmaniasis, with IRS playing a crucial role in reducing seasonal *P. argentipes* populations [2,13,18]. Ensuring the continued efficacy of IRS is of great importance, both through monitoring of application spray rates and of warning signs for an impact of insecticide resistance in the targeted *P. argentipes* vector. Programmatic use of pyrethroids for IRS creates an inherent vulnerability to the threat of resistance, due to a shared target site with DDT, to which resistance has become well established [6,8,13]. Resistance to alpha-cypermethrin, the pyrethroid used for IRS in India, has yet to be demonstrated and was not recorded in tests performed annually between 2016–2019

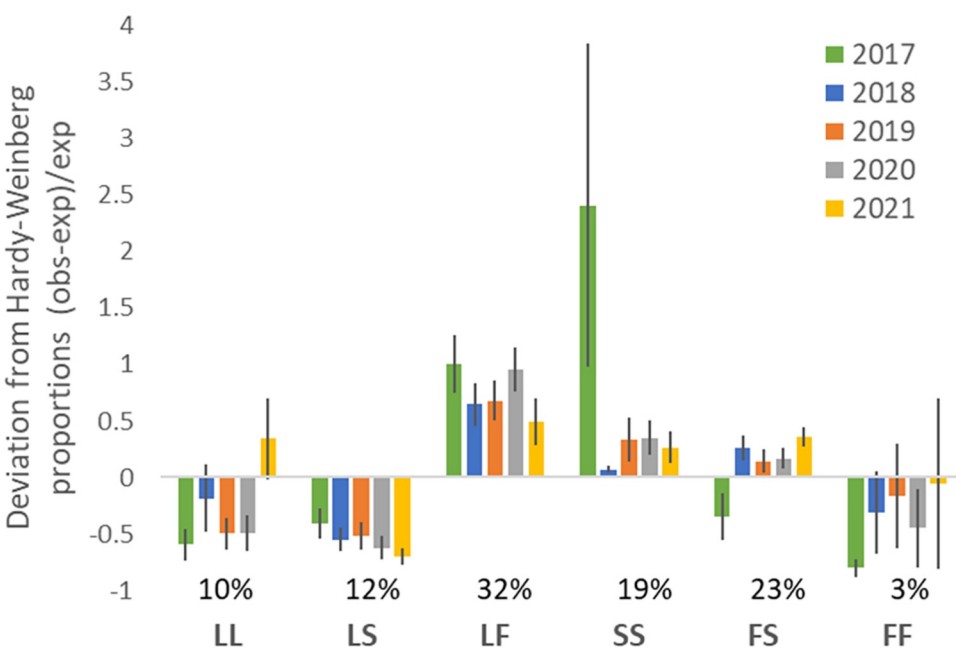

**Fig 6. Deviation of each 1014 genotype from Hardy-Weinberg expectations in each year of sampling (mean across districts +/- standard error).** Overall average percentage frequencies of each genotype are shown above genotype labels.

[13]. However, pyrethroid resistance measured using the same *Anopheles* bioassay thresholds, which for alpha-cypermethrin (though not deltamethrin) are lower than for *P. argentipes*,[19] has recently emerged in Nepal, especially to alpha-cypermethrin, and appears to be more common in villages receiving IRS [20]. Though this might be linked to an earlier switch to pyrethroids in Nepal [20] than India, and operational impacts remain unclear, these results clearly highlight the need for vigilance. By monitoring frequencies of the 1014 *kdr* mutations across the sentinel site system in north-eastern India across a five-year period we sought evidence for changes which could provide early indication of changing resistance profiles.

Results showed a significant increase from an average of approximately 35% *kdr* genotype frequency (i.e. those possessing two mutant alleles) to approximately 50% between 2017 and 2018, but thereafter little evidence of further increase in subsequent years, and a slight decline in 2021. The increase from 2017 levels was significantly more pronounced in areas with lower starting *kdr* genotype frequencies, most notably in the districts of Godda and Katihar which showed over four-fold increases. Nevertheless, spatial variation–primarily evident among, rather than within districts was pronounced in 2017, and remained so across the five study years. Frequency of *kdr* showed an imperfect longitudinal pattern with eastern sample sites tending to be lower and the three high *kdr* sites, Gopalganj, Muzaffarpur, and Samastipur, more western, though the western site East Champaran exhibited relatively low *kdr*, precluding simple geographical interpretation of patterns. Relatively high *kdr* frequencies (≈60% 1014F and S alleles) have been reported previously in West Bengal [10]. Whilst like the overall average we detected, this is much higher than in our West Bengal site, Darjeeling, though Sardar et al.'s study also included additional more southern sample sites [10]. Very high frequencies of *kdr* mutants have also been recorded in Bangladesh, further challenging the idea of simple geographical patterns. Indeed, near fixation of *kdr* alleles in both *P. argentipes* and another phlebotomine *Sergentomyia babu babu* were found in Myrmensingh, which had received prolonged IRS treatment, with much lower frequency in Pabna district which has a much shorter

history of IRS [21]. The primary difference between sites was a much higher frequency of the 1014F mutation in Myrmensingh [21], which contrasts with a fourfold lower frequency in collection made several years earlier from the same district [21].

Our results also provided evidence for links between IRS coverage, in terms of the proportion of houses sprayed or when comparing IRS and non-IRS villages (in 2017), as well as proportionate under spraying. An elevated frequency of the 1014F was consistently positively associated with IRS in each test, whilst association of specific genotypes varied. Association of 1014F is consistent with the results from Bangladesh above [21], but in our results this did not necessarily translate into high frequencies of the 1014FF homozygote genotype expected to cause the most resistant phenotype. In fact, this genotype was consistently under-represented in the dataset compared to Hardy-Weinberg expectations, with relative over-representation of 1014LF heterozygotes. Moreover, given the high IRS pressure, the stability of *kdr* alleles from 2018 onwards is not consistent with the advantage implied by the positive IRS-*kdr* associations translating into consistent selective pressure.

There are several possible explanations for this. (1) The positive associations reflect past history of spraying with DDT, prior to the study period, rather than contemporary patterns. This is possible for the coverage data, owing to a very strong correlation between alpha-cypermethrin spray coverage and prior DDT spray coverage, however, it is less likely to generate associations between *kdr* allele frequencies in IRS vs non-IRS villages in 2017, and does not link with evidence for under spraying. (2) The 1014FF homozygote genotype confers substantial fitness costs. A study on different but nearby *kdr* mutants (S989P and V1016G) in *Ae. aegypti* [22] which backcrossed the mutants into an insecticide susceptible strain, and another of 1014F, which was introduced to a susceptible *An. gambiae* strain by genome editing [23], each documented strong fitness costs affecting both larval and adult stages. Whilst a strong candidate explanation, evidence from phlebotomines would be required to confirm this hypothesis. (3) The 1014LF heterozygote confers some resistance, perhaps balanced by a lower cost than 1014FF homozygotes. Laboratory data suggested that the 1014F and S mutants were largely recessive, although of the heterozygote wildtype/mutant genotypes, data for 1014LF were the most ambiguous with relatively closer frequencies in survivors and dead in deltamethrin assays [8]. Further investigation is required to confirm a possible advantage of this genotype, which if selected would maintain balanced frequencies of resistant and wild type alleles. (4) IRS is not selecting as strongly for *kdr* in north-eastern India, either because of differences in the spray programme, local ecology, or in the *P. argentipes* population. Consistent with this is the observation that 1014F frequency rarely exceeded 40% in any site-year sample in our dataset, whereas in Myrmensingh 1014F frequency exceeded 70% [21]. Population differences might involve additional mutations or mechanisms found in Bangladesh which interact with 1014F to elevate resistance or reduce costs. Such secondary non-synonymous variants are common in the Vgsc of *An. gambiae* [15] and *Ae. aegypti* [24] and might also involve interaction with variants in other genes beyond the Vgsc [25,26] To date Vgsc sequencing in *P. argentipes* that harbour any *kdr* mutations has been limited to a relatively short section flanking the 1014 codon, precluding current evaluation of this explanation.

Separating the above hypotheses will require additional studies, of which further association testing of different Vgsc 1014 genotypes would provide clarity on their association with resistance, preferably using more field-relevant assays, such as exposure to pyrethroid-sprayed surfaces [11]. Studies on relative fitness costs of 1014FF homozygotes in relation to 1014FS heterozygotes would also give insight into their potential cost-benefit balance. Sequencing of the whole Vgsc, from Indian and Bangladeshi populations to explore the contrasting presence of additional mutants could be especially valuable and could highlight additional markers for screening. In addition, exploration of additional resistance mechanisms, beyond the Vgsc,

which have been documented via broad-spectrum biochemical assays in *P. argentipes* [12,27] could inform of the sufficiency of Vgsc mutations to generate resistance phenotypes. Irrespective of the explanation, the positive associations we found between IRS and *kdr*, should serve as a cautionary note, especially for continued vigilance to maintain and improve spray quality, given the positive association between *kdr* and under spraying detected.

An acknowledged limitation of the study is the use of *kdr* genotype data as a proxy for phenotypic resistance. Whilst this presents significant logistical advantages and also may offer the potential for more sensitive detection of changes in resistance, testing for phenotypic resistance to insecticides; specifically, alphacypermethrin should be conducted. At large scale this can be difficult in *P. argentipes* due to challenges with rearing and, until very recently, a lack of diagnostic insecticide doses for susceptibility [14], but conclusions would have been strengthened by availability of results from alphacypermethrin bioassays. In addition, and once discovered and validated, perhaps more tractable at large scale, other resistance mechanisms could also be explored. For example variation in gene complement, sequence or expression of glutathione S-transferase (GST) genes appear promising candidates for studies of metabolic resistance in phlebotomies [27,28], and the role of other major detoxification gene families, perhaps most notably cytochrome P450s remains to be investigated.

## Conclusion

The overall results from the study are positive for the VL elimination programme, in that a significant increase in *kdr* resistance marker frequency from 2017 to 2018 did not continue, suggesting that further progress towards pyrethroid resistance did not occur during the 5-year monitoring period. However, there are warning signs that IRS links with relative advantage of certain genotypes which are more resistance-associated. Coupled with recent emergence of pyrethroid resistance in Nepal, this indicates that alternatives insecticides should be incorporated into an integrated resistance management strategy. The value of molecular surveillance for the VL programme will be improved by additional quantification of genotype-phenotype associations, preferably from more operationally relevant phenotypic monitoring, and investigation of additional resistance mechanisms.

## Supporting information

**S1 Table. Generalised linear model results for allele and genotype frequencies as predicted by district and year.**
(DOCX)

**S2 Table. Marker genotyping data for each *P. argentipes* individual included in the study.**
(XLSX)

**S3 Table. Marker data summarised by village including spray proportion metrics.**
(XLSX)

**S1 Fig. Scatterplot illustrating relationships between alpha-cypermethrin-IRS coverage index each year and prior DDT-IRS coverage in the same groups of villages.** A linear regression line is fitted based on a multi-year average of the alpha-cypermethrin-IRS data.
(TIF)

## Acknowledgments

We thank the National Vector Borne Disease Control Programme for facilitating this work. and the villages in which we have had the pleasure to work.

## Author Contributions

**Conceptualization:** Rinki Michelle Deb, Michael Coleman, David Weetman.

**Data curation:** Rinki Michelle Deb, Asgar Ali, Rudra Pratap Singh, Josephine Shepherd, Anand Mohan Singh, Aakanksha Bharti, Michael Coleman.

**Formal analysis:** Emma Reid, David Weetman.

**Investigation:** Rinki Michelle Deb, Asgar Ali, Rudra Pratap Singh, Josephine Shepherd, David Weetman.

**Methodology:** David Weetman.

**Project administration:** Chandramani Singh, Sadhana Sharma, Michael Coleman.

**Resources:** Prabhas Kumar Mishra, Michael Coleman.

**Supervision:** Rudra Pratap Singh, Prabhas Kumar Mishra, Chandramani Singh, Sadhana Sharma, Michael Coleman, David Weetman.

**Writing – original draft:** Emma Reid, Michael Coleman, David Weetman.

**Writing – review & editing:** Emma Reid, Rinki Michelle Deb, Asgar Ali, Rudra Pratap Singh, Michael Coleman, David Weetman.

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
