## [Decision Letter · Decision Letter 0]

18 Jul 2023

Dear Dr Coleman,

Thank you very much for submitting your manuscript "Molecular surveillance of insecticide resistance in Phlebotomus argentipes targeted by indoor residual spraying for visceral leishmaniasis elimination in India" for consideration at PLOS Neglected Tropical Diseases. As with all papers reviewed by the journal, your manuscript was reviewed by members of the editorial board and by several independent reviewers. The reviewers appreciated the attention to an important topic. Based on the reviews, we are likely to accept this manuscript for publication, providing that you modify the manuscript according to the review recommendations. 

Please address the reviewers concerns, especially clarifications requested by the second reviewer on phenotypic resistance to alphacypermethrin.

Sincerely,

Paul O. Mireji, PhD

Academic Editor

Nigel Beebe

Section Editor

Please address the reviewers concerns, especially clarifications requested by the second reviewer on phenotypic resistance to alphacypermethrin.

Reviewer's Responses to Questions

**Key Review Criteria Required for Acceptance?**

**Methods**

-Are the objectives of the study clearly articulated with a clear testable hypothesis stated?

-Is the study design appropriate to address the stated objectives?

-Is the population clearly described and appropriate for the hypothesis being tested?

-Is the sample size sufficient to ensure adequate power to address the hypothesis being tested?

-Were correct statistical analysis used to support conclusions?

-Are there concerns about ethical or regulatory requirements being met?

Reviewer #1: -Are the objectives of the study clearly articulated with a clear testable hypothesis stated? YES

-Is the study design appropriate to address the stated objectives? YES

-Is the population clearly described and appropriate for the hypothesis being tested? YES

-Is the sample size sufficient to ensure adequate power to address the hypothesis being tested? YES

-Were correct statistical analysis used to support conclusions? YES

-Are there concerns about ethical or regulatory requirements being met? NO

Reviewer #2: The objectives and study design are clear and well described. Statistical analysis has been designed to adress the oucomes. The sample size is appropriate (no of cluster, no of samples tesetd) to asses difficerences between treatments. No ethical concern was noted.

Reviewer #3: Methods are clearly described with appropriate statistical analysis to support the conclusion. The use of generalized linear models with appropriate covariates and random effects accounts for clustering and potential influences of IRS (indoor residual spraying) on genotype and allele frequencies. The described data analysis methodology seems appropriate for examining the hypothesis regarding changing resistance profiles and their potential links with the spraying program.

**Results**

-Does the analysis presented match the analysis plan?

-Are the results clearly and completely presented?

-Are the figures (Tables, Images) of sufficient quality for clarity?

Reviewer #1: -Does the analysis presented match the analysis plan? YES

-Are the results clearly and completely presented? YES

-Are the figures (Tables, Images) of sufficient quality for clarity? YES (though see minor comments below)

Reviewer #2: The results section is well structured and provide enough details to understand the findings. The english is correct.

The qulaity of the figure can however be improved

Reviewer #3: The authors should consider making improvements to the quality of the figures in order to enhance clarity in the study. They should consider converting several tables, such as Table 1 and 2, into graphical representations (graphs) to better present the data. This approach can make the information more visually appealing and easier to interpret for the readers.

**Conclusions**

-Are the conclusions supported by the data presented?

-Are the limitations of analysis clearly described?

-Do the authors discuss how these data can be helpful to advance our understanding of the topic under study?

-Is public health relevance addressed?

Reviewer #1: -Are the conclusions supported by the data presented? YES

-Are the limitations of analysis clearly described? YES

-Do the authors discuss how these data can be helpful to advance our understanding of the topic under study? YES

-Is public health relevance addressed? YES

Reviewer #2: The conclusions should be amended to adress referee's comments as described below.

Reviewer #3: The data presented in the study supports the conclusions drawn by the authors, indicating that further progress towards pyrethroid resistance did not occur during the monitoring period for visceral leishmaniasis elimination. The limitations of the analysis are not explicitly described, but the authors suggest the need for additional research on genotype-phenotype associations and other resistance mechanisms. They discuss the value of molecular surveillance for the VL program and emphasize the importance of incorporating alternative insecticides into an integrated resistance management strategy. The authors' conclusions provide practical implications for controlling insecticide resistance and advancing the VL elimination program.

**Editorial and Data Presentation Modifications?**

Reviewer #1: This is a clearly analysed and reported piece of work on an impressive amount of collected data. The methodology appears to be sound and my only comments are on minor typographical errors (listed below).

Line 85 - missing closing bracket and full stop.

Line 126-127 - there appears to be 9 rather than 8 sites listed. Is “East” in the wring place?

Line 130-131 - sentence needs altering (perhaps changing the comma to an ‘and’?)

Line 153 - an extra closing bracket needs removing.

Lines 174, 179 (and throughout) - perhaps use ‘GzLM’, to distinguish generalized from general linear model.

Line 414 - remove final “in”.

Figure 1 - would benefit from being inset in a larger map to indicate broader geographical location, and perhaps different shading of the districts studied.

Figure 4 - double-check East Champaran data (appears to show no change between 2017 and 2021).

Reviewer #2: (No Response)

Reviewer #3: (No Response)

**Summary and General Comments**

Reviewer #1: This is a clearly analysed and reported piece of work on an impressive amount of collected data.

Reviewer #2: Dear Editor,

The authors investigated the wide-scale spatio-temporal molecular variation of kdr mutations in Phlebotomus argentipes sampled from sentinel sites of north-eastern India during a period of intensive alpha-cypermethrin spraying for vector control. The idea was to assess whether changes in resistance marker frequency have occurred after the implementation of IRS with a pyrethoid. Overall, the authors found an increase in kdr genotype frequency between 2017 and 2018, but no significant changes in kdr composition afterward, with no evidence linking IRS activities to selection of any kdr genotypes (SS or FF). Despite the fact that the authors didn’t investigate the phenotypic resistance and the presence of resistance markers other than the Kdr, the paper provides very relevant information to guide decision making for vector control and resistance management.

Major Comment. The survey did not provide any information about the phenotypic resistance to alphacypermethrin in the sandflies collected during the period of treatment. Although I understand the complexity and difficulty to conduct bioassays according to WHO or CDC protocols, the authors don’t really know whether the overall level of resistance to alphacypermethrin (AC) has (or not) increased during the 4 years? Actually, the kdr was used as “a proxy” but other resistance mechanisms than kdr may be present and being under selection due to AC. This lack of information may be critical for the choice of insecticide to use for IRS in the coming years. Since the study focus on molecular markers only it would have been relevant to better explore the genome of the sandflies collected to assess potential regions/genes under selection using NGS and bioinformatic tools. This is particularly relevant since the kdr data did not provide outstanding results. The authors should also acknowledge this in the conclusion because it may be misleading to think that an absence of significant change in kdr may be sufficient to prevent alpha cypermethrin resistance to spread following successive rounds of IRS with this insecticide.

Reviewer #3: The authors offer valuable insights into insecticide resistance in Phlebotomus argentipes, the sand fly vector of visceral leishmaniasis (VL) in north-eastern India. Their use of molecular resistance assays to examine changing resistance profiles and their association with the spraying program is commendable. The findings, showing varying mutant frequencies and a significant change in the initial two years followed by stability, contribute to our understanding of insecticide resistance dynamics. The positive correlation between kdr alleles and spray coverage, as well as under-spraying, underscores the susceptibility to pyrethroid resistance development. However, the authors also note the infrequent detection of highly resistance-conferring mutant genotypes, indicating the absence of significant resistance despite widespread spraying. Although encouraging for the VL elimination program, the authors rightly suggest exploring alternative insecticides due to the advantage of resistance alleles in sprayed areas.

PLOS authors have the option to publish the peer review history of their article (what does this mean?). If published, this will include your full peer review and any attached files.

Reviewer #1: No

Reviewer #2: No

Reviewer #3: No

Figure Files:

Data Requirements:

Reproducibility:

References

---

## [Decision Letter · Decision Letter 1]

17 Oct 2023

Dear Dr Coleman,

We are pleased to inform you that your manuscript 'Molecular surveillance of insecticide resistance in Phlebotomus argentipes targeted by indoor residual spraying for visceral leishmaniasis elimination in India' has been provisionally accepted for publication in PLOS Neglected Tropical Diseases.

Best regards,

Paul O. Mireji, PhD

Academic Editor

Nigel Beebe

Section Editor

Reviewer's Responses to Questions

**Key Review Criteria Required for Acceptance?**

**Methods**

-Are the objectives of the study clearly articulated with a clear testable hypothesis stated?

-Is the study design appropriate to address the stated objectives?

-Is the population clearly described and appropriate for the hypothesis being tested?

-Is the sample size sufficient to ensure adequate power to address the hypothesis being tested?

-Were correct statistical analysis used to support conclusions?

-Are there concerns about ethical or regulatory requirements being met?

Reviewer #3: (No Response)

**Results**

-Does the analysis presented match the analysis plan?

-Are the results clearly and completely presented?

-Are the figures (Tables, Images) of sufficient quality for clarity?

Reviewer #3: (No Response)

**Conclusions**

-Are the conclusions supported by the data presented?

-Are the limitations of analysis clearly described?

-Do the authors discuss how these data can be helpful to advance our understanding of the topic under study?

-Is public health relevance addressed?

Reviewer #3: (No Response)

**Editorial and Data Presentation Modifications?**

Reviewer #3: (No Response)

**Summary and General Comments**

Reviewer #3: (No Response)

PLOS authors have the option to publish the peer review history of their article (what does this mean?). If published, this will include your full peer review and any attached files.

Reviewer #3: No

---

## [Editor Report · Acceptance letter]

29 Oct 2023

Dear Dr Coleman,

We are delighted to inform you that your manuscript, "Molecular surveillance of insecticide resistance in Phlebotomus argentipes targeted by indoor residual spraying for visceral leishmaniasis elimination in India," has been formally accepted for publication in PLOS Neglected Tropical Diseases.

Best regards,

Shaden Kamhawi

co-Editor-in-Chief

Paul Brindley

co-Editor-in-Chief
